# A Comprehensive Evaluation of Gridded L-, C-, and X-Band Microwave Soil Moisture Product over the CZO in the Central Ganga Plains, India

**Saroj Kumar Dash** 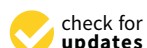 and **Rajiv Sinha** *

Department of Earth Sciences, Indian Institute of Technology Kanpur, Kanpur 208016, India; saroj@iitk.ac.in
* Correspondence: rsinha@iitk.ac.in

**Abstract:** Recent developments in passive microwave remote sensing have provided an effective tool for monitoring global soil moisture (SM) observations on a spatiotemporal basis, filling the gap of uneven in-situ measurement distribution. In this paper, four passive microwave SM products from three bands (L, C, and X) are evaluated using in-situ observations, over a dry–wet cycle agricultural (mostly paddy/wheat cycle crops) critical zone observatory (CZO) in the Central Ganga basin, India. The L-band and C/X-band information from Soil Moisture Active Passive (SMAP) Passive Enhanced Level 3 (SMAP-L3) and Advanced Microwave Scanning Radiometer 2 (AMSR2), respectively, was selected for the evaluation. The AMSR2 SM products used here were derived using the Land Parameter Retrieval Model (LPRM) algorithm. Spatially averaged observations from 20 in-situ distributed locations were initially calibrated with a single and continuous monitoring station to obtain long-term ground-based data. Furthermore, several statistical metrices along with the triple collocation (TC) error model were used to evaluate the overall accuracy and random error variance of the remote sensing products. The results indicated an overall superior performance of SMAP-L3 with a slight dry bias ($-0.040$ m$^3$·m$^{-3}$) and a correlation of 0.712 with in-situ observations. This also met the accuracy requirement (0.04 m$^3$·m$^{-3}$) during most seasons with a modest accuracy (0.059 m$^3$·m$^{-3}$) for the entire experimental period. Among the LPRM datasets, C1 and C2 products behaved similarly ($R = 0.621$) with a ubRMSE of 0.068 and 0.081, respectively. The X-band product showed a relatively poor performance compared to the other LPRM products. Seasonal performance analysis revealed a higher correlation for all the satellite SM products during monsoon season, indicating a strong seasonality of precipitation. The TC analysis indicated the lowest error variance ($0.02 \pm 0.003$ m$^3$·m$^{-3}$) for the SMAP-L3. In the end, we introduced Spearman's rank correlation to assess the dynamic response of SM observations to climatic and vegetation parameters.

**Keywords:** soil moisture; critical zone observatory; SMAP; AMSR2; evaluation; triple collocation

## 1. Introduction

Soil moisture (SM) is a key variable of the Earth system, influencing the hydrological cycle through evaporation and surface energy fluxes. Accurate and timely observation of SM at high spatiotemporal scale is of great importance in understanding surface hydrological processes, agricultural applications, hydrometeorological monitoring, and climate change impacts [1–4]. Although in-situ measurements by gravimetric and electromagnetic sources provide precise and accurate information on the SM variability, it is practically difficult to assess long-term variability in high-spatial-resolution because of the uneven distribution and uncertainties in the monitoring network.

Recent developments in microwave remote sensing, especially the L-, C-, and X-bands, have been widely used for global or regional SM mapping [5,6]. The C-band (6.9 GHz) and X-band (10.7 GHz) data from Advanced Microwave Scanning Radiometer-Earth Observing System (AMSR-E) on board the National Aeronautics and Space Administration (NASA)

Aqua satellite, launched in 2002, provided the first global microwave satellite SM [7]. However, a problem in the antenna rotation led to the launch of its successor, the Advanced Microwave Scanning Radiometer 2 (AMSR2) on board the Global Change Observation Mission First Water (GCOM-W1) of the Japan Aerospace Exploration Agency (JAXA) in 2012. Moreover, a new C-band (7.3 GHz) was added to the AMSR2 because of the radiofrequency interference observed at the C-band of AMSR-E [8,9]. On the other hand, the L-band was found to be optimal for the surface soil moisture monitoring because of its higher sensitivity toward vegetation and soil penetration than higher frequencies [7].

The spaceborne gridded SM data provide a unique opportunity for various land applications such as weather/climate forecasting [10], flood modeling and prediction [11–13], drought monitoring [14], and crop yield [15]. While the regional application of the SM dataset is necessary, there are significant challenges in the evaluation of the current satellite products because of the spatial mismatch of ground-based observations with large spatial heterogeneity [7,16,17]. Despite the challenges, numerous studies have documented the importance of in-situ point-based monitoring networks for satellite and reanalysis soil moisture validation experiments in various geographical regions [16,18–25]. For instance, an assessment of 9 km resolution L-band level-3 and level-4 product from Soil Moisture Active Passive (SMAP) in the United States [26] demonstrated a strong agreement of the level-4 SM with in-situ observations. The core validation sites for the SMAP calibration/validation experiment provided an excellent outcome in the performance of different SMAP radiometer products [27]. Over the Tibetan Plateau, evaluation of the level-3 enhanced SMAP product revealed a stronger correlation (0.65–0.88) with a slight larger unbiased root-mean-square error (ubRMSE) variability (0.055 to 0.059), exceeding the error requirement for the SMAP mission [28], while the regional SM showed an underestimation of SMAP radiometer level-3 global daily SM product with a negative bias [29]. A recent study [30] showed a moderate correlation with a ubRMSE of 0.055 $m^3 \cdot m^{-3}$ for the descending SMAP level-3 enhanced SM product. In addition to L-band SM evaluation, many studies have evaluated the potential of the AMSR-E and AMSR2 (C/X-band) SM products toward ground SM characterization. The SM products from AMSR2 were retrieved using two algorithms: the Japan Aerospace Exploration Agency (JAXA) algorithm [31] and the land parameter retrieval model (LPRM) [32,33]. Studies revealed a better performance of the LPRM compared to the JAXA algorithm when considering the bias and root-mean-square error [34]. In contrast, a better performance of JAXA products than LPRM was suggested while evaluating the core validation sites around the world [35]. The authors also suggested that the difference in sensing depth between in-situ sensor and satellite might cause the potential error. Various studies have proposed the combined validation of multiple satellite SM data to depict a comparable accuracy. A comparable study of eight satellite SM data including SMAP and AMSR2 demonstrated an outperformance of SMAP with respect to AMSR2 with an ubRMSE of 0.027 $m^3 \cdot m^{-3}$ in United States [19]. Two studies [19,36] reported an overestimation with a wet bias (0.09 $m^3 \cdot m^{-3}$) for the LPRM AMSR2 product in Little Washita Watershed (LWW), United States and in the global validation network respectively.

In addition to the above SM validation studies, efforts have also been made to utilize the in-situ datasets for validation of coarser remote sensing SM products through downscaling using auxiliary variables. There are numerous SM downscaling techniques that extract information from ancillary datasets in a way to combine the coarser passive or active microwave products with the finer visible or thermal infrared auxiliary products [37,38], including the geographic information along with topography [39]. Furthermore, several statistical and physical models have been previously developed for SM downscaling [40]. Additionally, the smoothening filter-based modulation (SFIM) technique uses the auxiliary information from the same instrument to produce the downscaled brightness temperature, which is used further in providing the AMSR2 SM through LPRM [33,41]. A comprehensive review of the downscaling protocols was recently published [42], and the assessment of various downscaling SM products has been documented globally [43]. However, several uncertainties exist in these products such as significant errors over complex terrains, opti-

mization of the use of several auxiliary datasets, and misinterpretation of the land surface models [39] resulting in significant errors.

An additional error model for random error variance in satellite SM evaluation has been used in several studies along with the above statistical metrics. The error can be quantified spatiotemporally by introducing a third independent dataset of the same geophysical parameter, referred to as triple collocation (TC) error [44–46]. A better performance of SMAP than AMSR2 was documented [25] when applying the TC analysis to in-situ data over the USA. A better accuracy of SMAP was also observed over multiple regions across the globe using the TC error model when compared with other satellite products [47]. A similar analysis of the L-band SM products was also reported by other workers [26,48].

A superior performance of the L-band SM products has not only been noticed globally [30,36] but has also been highlighted recently for many geographical regions of India [47,49–51]. However, very few studies have addressed the remote sensing-based SM validation over the croplands, especially in the cyclic (paddy and wheat)-dominated regions, which are most prominent in the Asian countries. The Ganga basin produces 50% of the India's total food grain and is, hence, known as the "food basket" of India. Recognizing the lack of ground-based observational data and the heterogeneity in the Ganga basin, especially the central Ganga plains, a critical zone observatory (CZO) was established to monitor the multiple hydrological observations and water resource management strategies [52]. The CZO in Ganga basin is characterized as an agricultural (mostly rice/wheat) land with frequent wet and dry cropping patterns. The study region is lacking any evaluation of the recent satellite-based soil moisture products. Given the importance of the temporal variability of soil moisture in the cropping patterns and anthropogenic inputs, the validation of the current satellite SM products has significant implications for the assessment of crop water requirements and hydrological modeling.

Therefore, the primary objective of this study is to evaluate the recently developed SMAP (L-band) and AMSR2 (C1, C2 and X-band) microwave SM products over the CZO by analyzing several statistical metrices along with the TC error model using the ground-based observations. The secondary objective is to assess the performance of the satellite and in-situ SM with the daily observed climatic and vegetation parameters.

## 2. Materials and Methods

### 2.1. Study Region

The validation of satellite datasets was conducted within the recently established critical zone observatory (CZO) in the Pandu River basin, which is a small plain-fed sub-basin of the river Ganga, north India. The areal extent of the CZO is approximately 22 km² with elevation varying from 126–143 m above sea level. The study site experiences a sub-humid climate regime with summer extremes of ~42 °C and winter extremes of ~9 °C. The area receives rainfall mostly in the monsoon (June–September) period with a mean annual rainfall of about 822 mm. The predominant soil type in this region is sandy loam to loam. This region is one of the rural landmarks of north India with agriculture as the major land-use pattern. Therefore, the CZO considered here has been designated as HEART (Heterogeneous Ecosystem of an Agro Rural Terrain) in the Ganga basin [52].

### 2.2. Remote Sensing Data

#### 2.2.1. SMAP

Launched by NASA on 31 January 2015, the SMAP satellite carries the payload to obtain global soil moisture and freeze/thaw state [27,53]. The SMAP is in the sun-synchronous near-polar orbit, providing soil moisture data from the top 0–5 cm of soil layer. The satellite carries an L-band radiometer (1.41 GHz) which gives the brightness temperature, in turn producing the soil moisture with a native resolution of 40 km. Soil moisture data are retrieved from both the descending and the ascending overpass at approximately 6:00 a.m. and 6:00 p.m. local time, respectively, with a revisit time of 2–3 days [27,53,54].

In this paper, we used the SMAP Level-3 Radiometer Global Daily 9 km enhanced product posted on the EASE-Grid (SMAP L3_SM_P_E, Version 3, hereafter referred to as SMAP-L3) [55] for validation against the in-situ observations over the CZO region (Figure 1). The SMAP radiometer data were acquired from the Earth Data Search portal of NASA (https://search.earthdata.nasa.gov/, accessed on 20 December 2021). The enhanced L3 soil moisture data represent a daily composite from the enhanced L2 product. The SMAP L2 product is derived from the SMAP L1C interpolated brightness temperature using the Backus–Gilbert optimal interpolation technique that extracts maximum information from the SMAP antenna temperatures [55–57]. In addition, the SMAP uses the V-pol single-channel algorithm (SCA-V) for the soil moisture estimation using vegetation optical depth (VOD) derived from vegetation water content. Here, we employed the VOD from the SCA-V and dual channel to examine the relationship between soil moisture and vegetation indices. Furthermore, we chose the surface soil moisture product from the descending (6:00 a.m.) overpass because of the diurnal frequency in the point data collection. In addition, the descending soil moisture product has the following advantages from a validation perspective: (1) soil and canopy temperature are consistent during the early morning; (2) temperature and the dielectric of the soil profile are likely to be uniform in the morning time; (3) the descending retrieval is more accurate than the ascending retrieval [54,56].

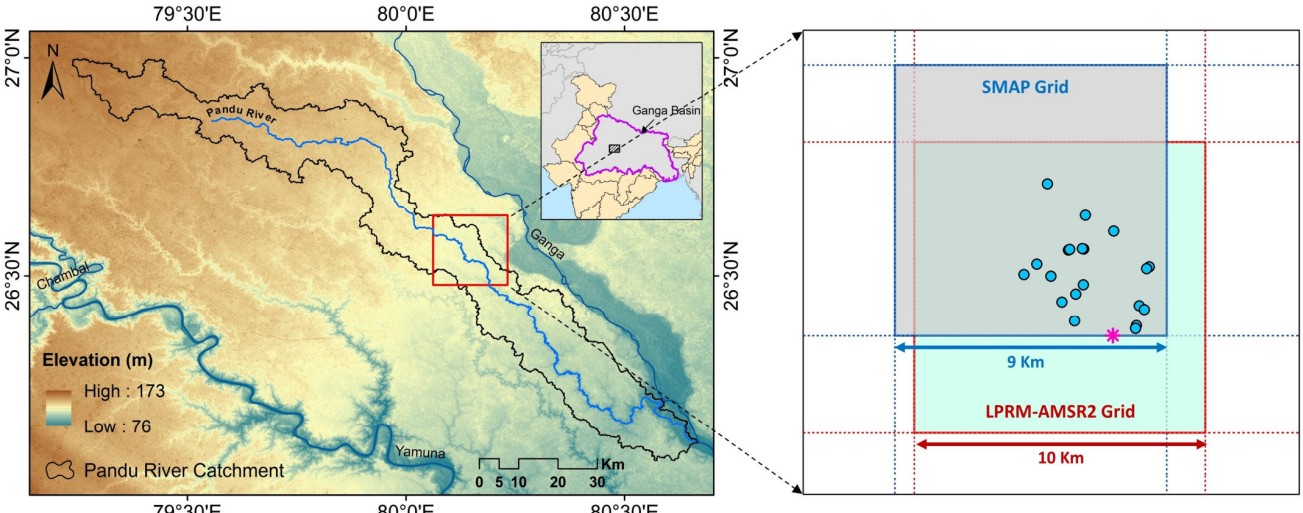

**Figure 1.** Location of the study area. The in-situ and satellite grid over the CZO sites in Ganga basin, north India. The blue filled circles indicate the ThetaProbe measurement locations, and the pink star shows the location of the automatic weather station downstream of the CZO, where the continuous soil moisture measurement is being monitored using a WaterScout SM100 sensor.

### 2.2.2. LPRM AMSR2

Onboard the Global Change Observation Mission 1-Water (GCOM-W1), AMSR2 is a passive microwave Earth observing satellite for global retrieval of soil moisture. It was launched by the Japan Aerospace Exploration Agency (JAXA) in May 2012. This is a successor to the previous AMSR-E, operated between May 2002 and October 2011.

The AMSR2 soil moisture comprises the 25 km grid resolution globally available ascending (1:30 p.m. local time) and descending (1:30 a.m. local time) data products. However, this study evaluates a recent release of the LPRM-based downscaled (10 km) soil moisture product version-1 (LPRM_AMSR2_DS_D_SOILM3) for the descending AMSR2 overpass [58]. The SFIM-based downscaling techniques are applied to the AMSR2 products, and the downscaled brightness temperatures are released and converted to SM using LPRM analysis [33,41]. Despite a reasonable spatial footprint, very few studies have explored the usefulness of the downscaled product [59,60]. Therefore, the present study aims to assess the corresponding LPRM AMSR2 (hereafter referred to as AMSR2) downscaled grid over

the study site for several hydrological applications. The spatial footprint of the AMSR2 is shown in Figure 1. The descending overpass was chosen because of the consistency of soil water content and the temperature, along with the timing of the manually measured ground data. Furthermore, this study used AMSR2 soil moisture and VOD derived from all three available bands (C1—6.9 GHz, C2—7.3 GHz, and X—10.7 GHz) for the evaluation strategy.

2.2.3. Moderate Resolution Imaging Spectroradiometer (MODIS) Vegetation Indices

The MODIS sensors are onboard two NASA satellites, Terra and Aqua, with 36 discrete spectral bands ranging in wavelength from 0.4 to 14.4 μm with a spatial resolution from 250 m to 1 km. In this study, the 16-day composite Terra MODIS vegetation indices, MOD13A2 version 006 [61] used for the parameter extraction, were downloaded from NASA Earth data search portal (https://search.earthdata.nasa.gov/, accessed on 20 December 2021). The original Hierarchical Data Format Earth Observing System (HDF-EOS) data were converted and projected to match the satellite soil moisture data format using HDF-EOS to GeoTIFF Conversion (HEG) Tool [62]. The MOD13A2 data were used to derive the normalized vegetation index (NDVI) and enhanced vegetation index (EVI), representative of 1 km spatial resolution from the acquisitions over the 16-day composite period. Overall, a total of 38 cloud-free NDVI and EVI images were used in this study to establish the correlation with the satellite-based and ground soil moisture evolution. The original data was resampled to SMAP and AMSR2 pixel resolution using the nearest neighbor resampling on a GIS platform.

The summary and overview of the remote satellite datasets used in this study are listed in Table 1.

**Table 1.** Overview and summary of the remote sensing products used in this study.

| Dataset | Variables | Spatial Coverage/Resolution | Period | Temporal Resolution/Local Overpass Time | Reference |
|---|---|---|---|---|---|
| **SMAP L3_SM_P_E** | Surface soil moisture, vegetation optical depth | 85.044°S to 85.044°N and 180°W to 180°E, 9 km | 31 March 2015 to present | 2–3 days, Descending—6:00 a.m., Ascending—6:00 p.m. | [55] |
| **LPRM AMSR2 DS D L3** | Surface soil moisture, vegetation optical depth | 90°S to 90°N and 180°W to 180°E, 10 km | 3 July 2012 to present | 1 day, Descending—1:30 a.m., Ascending—1:30 p.m. | [58] |
| **MOD13A2** | NDVI, EVI | Global, 1 km | 18 February 2000 to present | 16 days, Descending—10:30 a.m. (Terra) | [61] |

*2.3. In-Situ Observations*

In-situ soil moisture measurements were carried out with two modes of measurements. The manual mode of measurement was conducted at 20 agricultural sub-plots distributed throughout the CZO, whereas the continuous mode of observation was conducted at a single location (Figure 1). The manual mode of observation was facilitated by a handheld impedance-based ThetaProbe ML3 soil moisture sensor (Delta-T Devices, Cambridge, UK) which records the data down to 6 cm of the ground. For each measurement at the field site, we acquired the data through a five-vertex measurement plan towards the center of each sampling site and estimated the mean soil moisture value afterward. Furthermore, the ThetaProbe measurements were carried out for the sampling days mostly coinciding with the SMAP satellite descending overpass over the study region. This excludes the dry period (mostly May–June) when the measurements were difficult because of the hardened soil surface. The automatic and continuous modes of SM observations were carried out using the WaterScout SM100 sensor (Spectrum Technologies, Plainfield, IL, USA) (Figure 2).

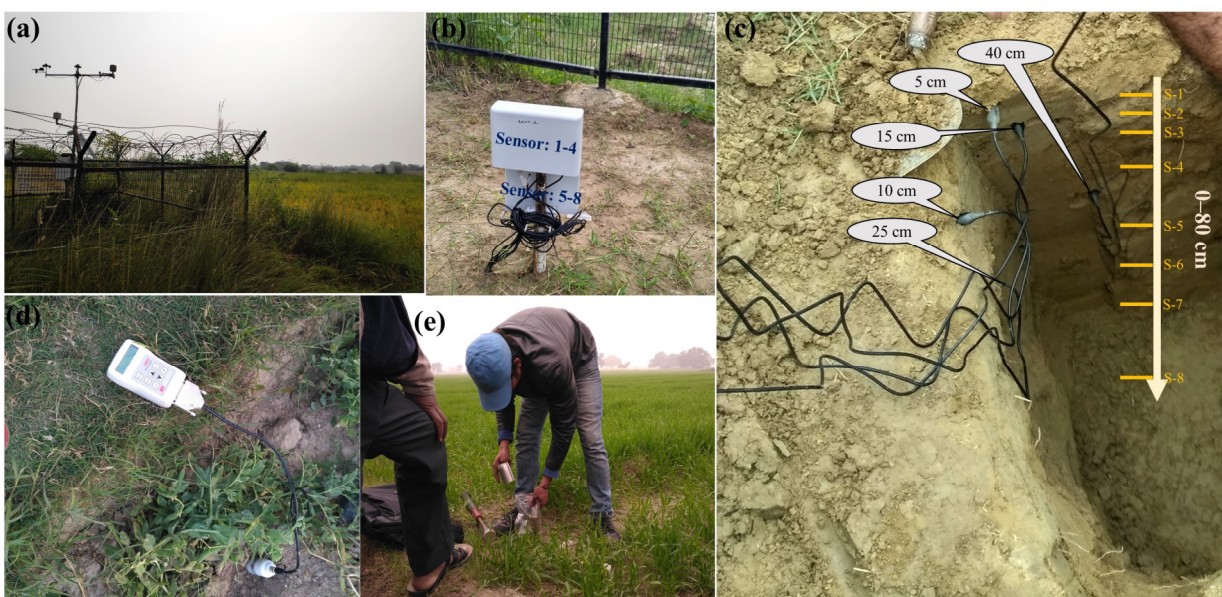

**Figure 2.** Field photographs showing (**a**) automatic weather station (AWS) downstream of the CZO, (**b**) WaterScout SM100 continuous soil moisture (SM) monitoring station at the AWS site acquiring data at eight subsurface level. Sensor: 1–4 indicates the first four sensor and Sensor: 5–8 indicates the next four sensors, (**c**) SM sensor installation locations at the subsurface, where S1–S8 indicate the sensor number at the corresponding locations, (**d**) SM measurement using the ML3 ThetaProbe sensor, and (**e**) collection of field soil sample for the gravimetric analysis at the laboratory.

The continuous in-situ soil moisture observation was acquired through the WaterScout SM100 (hereafter referred to as SM100) sensor, installed horizontally at different depths ranging from 0 to 80 cm for surface and subsurface volumetric water content observations (Figure 2). All surface and subsurface installed sensors were connected to the WatchDog 1000 series Micro Station data logger to record the continuous soil moisture data at 15 min intervals. The sampling period in this study covered a total 62 days for the manual measurements and 705 days for the SM100 continuous measurements during September 2017 and December 2019. Furthermore, this study used only the surface (5 cm) soil moisture of SM100 sensor for the diurnal temporal variation from 4:30 a.m. to 6:30 a.m. to coincide with the descending overpass time of both the SMAP and the AMSR2 satellites.

To estimate the accurate ground-based soil moisture values, both ML3 ThetaProbe and SM100 sensor were initially calibrated, considering the local soil types. Both sensors were calibrated using the gravimetric soil moisture measurements. The calibration was carried out with the assessment of volumetric soil water content using the sensor-based measurement and the gravimetric weight-based measurement of the soil samples. Initially, the measurement was carried out in-situ along with simultaneous collection of the soil sample using the soil sampler (Figure 2). The samples were then processed for volumetric water content measurement using the oven-dry method. Accordingly, 37 soil samples collected from different locations were used for gravimetric measurement of soil moisture and to derive the regression between the ML3 ThetaProbe and actual observations. However, it is worth mentioning that 20 samples are adequate to determine the standard error [63]. A regression was developed for the calibration of the SM100 sensor using conventional laboratory-based gravimetric measurement of 57 soil samples from a saturated state to the wilting state to capture the diurnal variation of soil moisture values.

*2.4. Evaluation Strategies*

Evaluation of a grid or pixel of the SM satellite product using the in-situ measurement a point location becomes challenging. The evaluation process in this study was fully based on the temporal data as the in-situ measurements in this study covered only a single model

grid for both the SMAP and the AMSR2 SM products (Figure 1). Since the in-situ SM ThetaProbe measurements (weekly) in the grid cell were temporally limited because of the seasonal variability of the surface soil properties (dry/wet), only SM100 measurements were considered for the validation process. However, the ThetaProbe measurement sites were considered to estimate the mean SM of the region and for ground-truthing, as they were spatially distributed throughout the satellite SM grid. Simple arithmetic averages of the observations from all sites were used to represent the in-situ SM value for the evaluation of the satellite grid. Furthermore, a regression between the CZO mean SM from the ThetaProbe locations and the corresponding SM100 measurements was explored, and a calibration between the two was subsequently established. Use of a single site for estimating the spatial mean SM through linear regression has been documented in many studies [64–66], and the catchment mean SM could be interpreted using the offsets derived from the linear regression [66]. In addition, selecting a single SM site randomly in the present catchment area led to a satisfactory relation ($R^2$ = 0.57) with the present catchment mean SM as estimated by the bootstrap resampling technique [67]. Therefore, the derived regression equation here was used for all the SM100 measurement to approximate the CZO mean SM value, representing the ground truth for both the SMAP and the AMSR2 SM grid.

2.4.1. Performance Metrics

The evaluation of the SMAP and AMSR2 satellite SM products was carried out using four statistical parameters—bias, root-mean-square error (RMSE), unbiased root-mean-square error (ubRMSE), and correlation coefficient (R). The bias represents the systematic difference between the satellite and ground measurement. Since the spatial resolution of the SMAP and AMSR2 are different, this might generate a biased RMSE [26,28,68]. Therefore, the ubRMSE is used to remove the bias to understand the error. The relative accuracy between the satellite SM product and the in-situ measurements is defined by the correlation coefficient, R.

$$\text{Bias} = \frac{1}{N}\sum_{i=1}^{N}\theta_{est}(i) - \frac{1}{N}\sum_{i=1}^{N}\theta_{true}(i),$$

$$\text{RMSE} = \sqrt{\frac{1}{N}\sum_{i=1}^{N}[\theta_{est}(i) - \theta_{true}(i)]^2},$$

$$\text{ubRMSE} = \sqrt{RMSE^2 - Bias^2},$$

$$\text{R} = \frac{\frac{1}{N}\sum_{i=1}^{N}\left[\left(\theta_{est}(i) - \frac{1}{N}\sum_{i=1}^{N}\theta_{est}(i)\right) \times \left(\theta_{true}(i) - \frac{1}{N}\sum_{i=1}^{N}\theta_{true}(i)\right)\right]}{\sigma_{est} \times \sigma_{true}},$$

where $N$ is the total number of sampling days of SM measurement, and $\theta_{est}$ and $\theta_{true}$ are the estimated (satellite) and observed (in-situ) soil moisture, respectively.

Additionally, Taylor diagrams [69] were used to describe the geometrical relationships of the three statistical parameters on a two-dimensional polar plot. This single diagram expresses the comprehensive visualization on the closeness of the two datasets as a function of their correlation, centered root-mean-square deviation (RMSD), and standard deviation. Moreover, it is worth mentioning that a direct comparison between the SMAP with AMSR2 SM product is somewhat meaningless as the remotely sensed footprint has different spatial resolution [26].

2.4.2. Triple Collocation (TC) Error Analysis

In this paper, we implemented the triple collocation error (TCA) analysis to estimate the random error variance in the SMAP, AMSR2, and in-situ datasets. The TCA model was initially proposed [45] for estimating the calibration constants within a satellite ocean wind product against reference data, and it has been popular in past years for soil moisture validation studies [25,26,44,47,70]. Three major assumptions exist in the application of TCA model: (1) three collocated datasets must be significantly correlated, i.e., the dataset must

represent the same physical quantity, (2) the covariances must be zero, i.e., independent error structures, and (3) an appropriate number of triplets must be available to obtain reliable estimates in the averaging step, i.e., sufficient triplets to estimate the error characteristics (triplets should be more than 100 [46]). Details on the assumptions along with the possible violations in applying the TCA were described by earlier workers [70,71]. In this study, we used the pytesmo python module to estimate the TCA [72]. The basic structure for estimating the random error variances through the TCA error model is described below.

Assuming three collocated independent SM datasets $\theta_1$ (SM100), $\theta_2$ (SMAP SM), and $\theta_3$ (AMSR2 SM), containing N observations, the relationship of the error structure and the hypothetical truth value can be defined for each as follows:

$$\theta_1 = \alpha_1 + \beta_1\theta + \varepsilon_1, \tag{1}$$

$$\theta_2 = \alpha_2 + \beta_2\theta + \varepsilon_2, \tag{2}$$

$$\theta_3 = \alpha_3 + \beta_3\theta + \varepsilon_3, \tag{3}$$

where $\theta$ is the true soil moisture, $\alpha_i$ and $\beta_i$ are the linear coefficients to the truth, and $\varepsilon_i$ is the random error to the corresponding $i = 1, 2, 3$. The major goal of TCA is to estimate the variances of $\varepsilon_1$, $\varepsilon_2$, and $\varepsilon_3$ [44], which can provide the quality of the datasets. The error variances can be resolved by differentiation of the covariances of the datasets [70].

$$\sigma^2_{\varepsilon_1\theta_1} = \sigma^2_{\theta_1} - \frac{\sigma_{\theta_1\theta_2}\sigma_{\theta_1\theta_3}}{\sigma_{\theta_2\theta_3}}, \tag{4}$$

$$\sigma^2_{\varepsilon_2\theta_2} = \sigma^2_{\theta_2} - \frac{\sigma_{\theta_2\theta_1}\sigma_{\theta_2\theta_3}}{\sigma_{\theta_1\theta_3}}, \tag{5}$$

$$\sigma^2_{\varepsilon_3\theta_3} = \sigma^2_{\theta_3} - \frac{\sigma_{\theta_3\theta_2}\sigma_{\theta_3\theta_1}}{\sigma_{\theta_2\theta_1}}, \tag{6}$$

where $\sigma^2_{\varepsilon_1\theta_1}$, $\sigma^2_{\varepsilon_2\theta_2}$, and $\sigma^2_{\varepsilon_3\theta_3}$ are the random error variances of the datasets $\theta_1$, $\theta_2$, and $\theta_3$ respectively. $\sigma^2_{\theta_1}$, $\sigma^2_{\theta_2}$ and $\sigma^2_{\theta_3}$ are the variances of the datasets $\theta_1$, $\theta_2$, and $\theta_3$ respectively. $\sigma_{ij}$ ($i$, $j = \theta_1$, $\theta_2$, $\theta_3$) represents the covariance of $i$ and $j$.

Additionally, the signal-to-noise ratio (SNR) for each dataset can be estimated by considering the variance and covariance of the triplets [70] as follows:

$$SNR_{\theta_1} = -10\log\left(\frac{\sigma^2_{\theta_1}\sigma_{\theta_2\theta_3}}{\sigma_{\theta_1\theta_2}\sigma_{\theta_1\theta_3}} - 1\right), \tag{7}$$

$$SNR_{\theta_2} = -10\log\left(\frac{\sigma^2_{\theta_2}\sigma_{\theta_1\theta_3}}{\sigma_{\theta_2\theta_1}\sigma_{\theta_2\theta_3}} - 1\right), \tag{8}$$

$$SNR_{\theta_3} = -10\log\left(\frac{\sigma^2_{\theta_3}\sigma_{\theta_1\theta_2}}{\sigma_{\theta_3\theta_1}\sigma_{\theta_3\theta_2}} - 1\right), \tag{9}$$

where $SNR_{\theta_1}$, $SNR_{\theta_2}$ and $SNR_{\theta_3}$ are the signal-to-noise ratios of the datasets $\theta_1$, $\theta_2$, and $\theta_3$, respectively. $\sigma^2_{\theta_1}$, $\sigma^2_{\theta_2}$, and $\sigma^2_{\theta_3}$ are the variances of the datasets $\theta_1$, $\theta_2$, and $\theta_3$, respectively. $\sigma_{ij}$ ($i$, $j = \theta_1, \theta_2, \theta_3$) represents the covariance of $i$ and $j$. The SNR here is expressed in decibels (dB) to make it symmetric around zero. Thus, an SNR value of zero denotes an equal variance of the signal and the noise, whereas a value of $\pm 3$ dB corresponds to doubling and halving of the ratio.

## 2.5. Spearman's Correlation Analysis

Nonparametric Spearman's rank correlation analysis was utilized in this paper to evaluate the interrelationship between in-situ and daily observed climatic variables (2 m air temperature, rainfall, and potential evapotranspiration (ET$_0$) with satellite and ground soil

moisture observations). The in-situ hydrometeorological observations were obtained from the automatic weather station installed at the field site (Figure 2). In addition, the FAO 56 Penman–Monteith [73] method was used to estimate the daily evapotranspiration using the daily observed climatic parameters such as temperature, relative humidity, wind speed, and solar radiation. Spearman's correlation was obtained based on the ranks determined for each variable under consideration, and the correlations of both ($r_s$) were estimated using the two ranked variables according to the following expression:

$$r_s = 1 - \frac{6 \sum_{i-1}^{n} \left( R_{X_i} - R_{Y_i} \right)^2}{n(n^2 - 1)},$$
(10)

where $R_{X_i}$ and $R_{Y_i}$ are the ranked variables of $X_i$ and $Y_i$ ($i = 1, 2, 3, \ldots, n$), respectively, and $n$ is the total number of elements for each variable. A rank correlation close to 1 indicates a stronger tendency of similarity between the variables.

## 3. Results

### 3.1. Relation between In-Situ Measurement Networks

Figure 3 presents the comparison of the measurements from soil moisture sensors and gravimetric method. A linear regression between the actual and measured soil moisture for both sensors showed extremely good correlation ($R^2$ = 0.951 for ThetaProbe and $R^2$ = 0.886 for SM100). The biases for the ThetaProbe and SM100 observations were found to be 0.51% ($v/v$) and $-0.8$% ($v/v$), respectively. An extremely good relationship between the actual and ThetaProbe SM was observed, whereas the SM100 observations were nonuniformly distributed, leading to clusters for the corresponding gravimetric measurements, although a pronounced high correlation was noted. The derived linear regression relations were successively used to estimate the accurate soil moisture content for the in-situ measurements from the successive field campaigns.

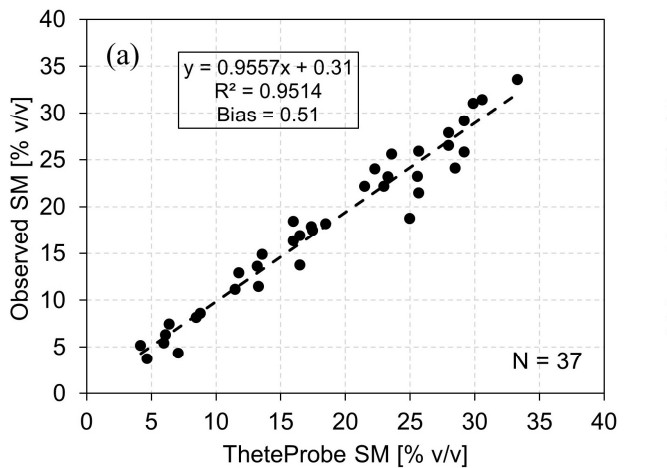
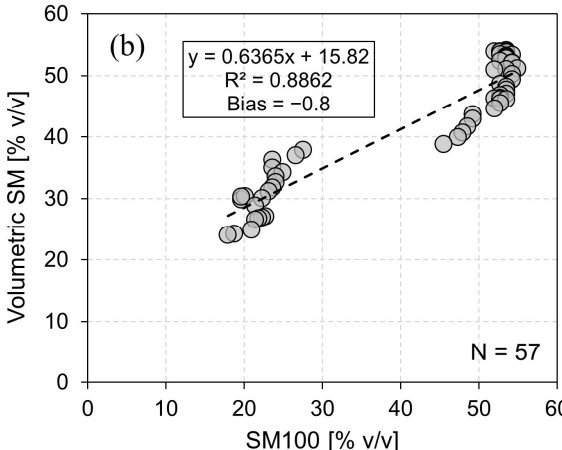

**Figure 3.** Comparison of the volumetric soil moisture measured using the sensor and the gravimetric procedure: (**a**) measurement for the manually collected sample using ML3 ThetaProbe; (**b**) measurement for the continuous observation of WaterScout SM100 sensor. The dotted line shows the linear regression between the observation and actual value. The calculated bias and correlation are shown for each comparison.

In this paper, we evaluated the SM100 observations against the satellite soil moisture product. Thus, we began by examining the CZO mean behavior of SM100 measurements with the spatially distributed ThetaProbe values. The linear regression was derived using the values coinciding with the same daily temporal measurement interval from both sensors, showing a mean bias of 0.06 m$^3$/m$^3$, correlation coefficient of 0.75, and $R^2$ of 0.56 ($p < 0.01$) (Figure 4). The linear relationship was then used to estimate the mean CZO soil

moisture from the daily SM observation from the SM100 sensor and successively used as the ground truth for the current evaluation study. Moreover, the bias became negligible ($-0.00003$ m$^3$/m$^3$) when considering the calibration for mean soil moisture estimation.

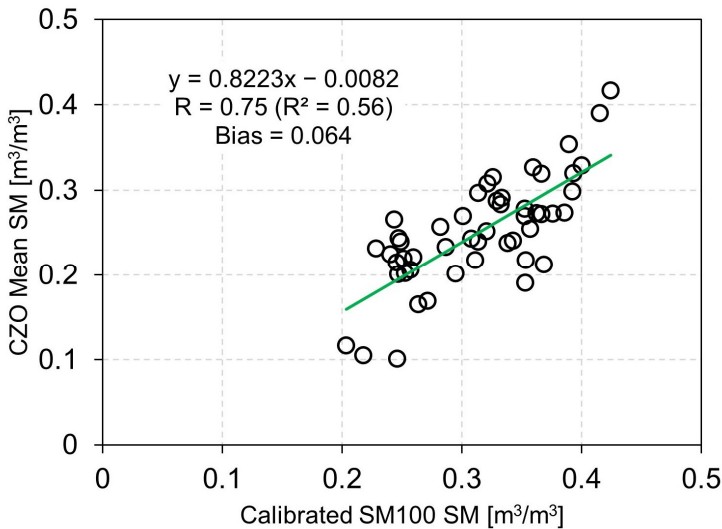

**Figure 4.** Scatter plot showing the relation between volumetric soil moisture from the continuous observation at the single location using the SM100 sensor and the manually collected mean catchment value using the ThetaProbe sensor around the CZO. The linear regressions are shown as a green line.

### 3.2. Timeseries and Seasonal Variability of Soil Moisture at CZO

Four satellite soil moisture products from SMAP and LPRM-AMSR2 were used here for validation using the in-situ data. Figure 5 shows the timeseries of the satellite soil moisture along with the ground based (ThetaProbe and CZO mean calibrated SM100) network for the entire study period. The complementary daily rainfall data were added to the timeseries to interpret the seasonal influence of SM–rainfall relationship. It can be clearly observed that both satellite and ground products strongly responded to the concurrent rainfall pattern with the maxima during monsoon months and minima during non-monsoon months. Furthermore, a constant SM value (0.48 m$^3$/m$^3$) was observed on the SMAP-L3 timeseries during the 2018 monsoon period (from 26 July 2018 to 12 September 2018). The constant SM value is indicative of the maximum SM retrieved by the SMAP-L3 product for the specific soil textures observed in the region. This threshold SM maximum was also observed on 26 July, 22 August, and 24 August 2019 for the SMAP-L3 timeseries. Soil moisture from the AMSR2 product grid showed a similar temporal variation for all three bands (C1-, C2-, and X-band).

The range of variability of soil moisture becomes noticeable when considering the seasonal annual pattern. The SM seasonal variation of SMAP-L3, AMSR2 (C1-, C2-, and X-band) is illustrated in Figure 6. The SM variability presented here was classified based on meteorological seasons in India: winter (January–February), pre-monsoon (March–May), monsoon (June–September), and post-monsoon (October–December). The range of SM variability for the SMAP L3 during winter was similar to the in-situ SM with median values of 0.22 m$^3$/m$^3$ and 0.24 m$^3$/m$^3$, respectively. A similar difference in the median soil moisture values between in-situ and SMAP L3 was noted during the monsoon period, when the annual rainfall was maximum. However, the dynamic range of SMAP L3 during this period was highest relative to the remaining periods for all SM products. It can also be observed that the AMSR2 satellite product presented a relatively higher range of variability compared to the in-situ observations. Furthermore, Figure 6 shows that the AMSR2-C1 product closely matched with the in-situ median soil moisture value during the pre-monsoon period, whereas the AMSR2-C2 better predicted the in-situ value during the monsoon period (0.28 m$^3$/m$^3$), although the range of variability for both was different. In general, the range of occurrence of 25–75% of all SM data was higher during the monsoon

period. This can be attributed to two reasons: (1) the disparity of the average annual rainfall pattern; (2) the monsoon period included here incorporated the month of June, which apparently falls under the summer dry period most of the time. The post-monsoon dynamics of the SM range for all satellite products is different and not in accordance with the in-situ SM observations.

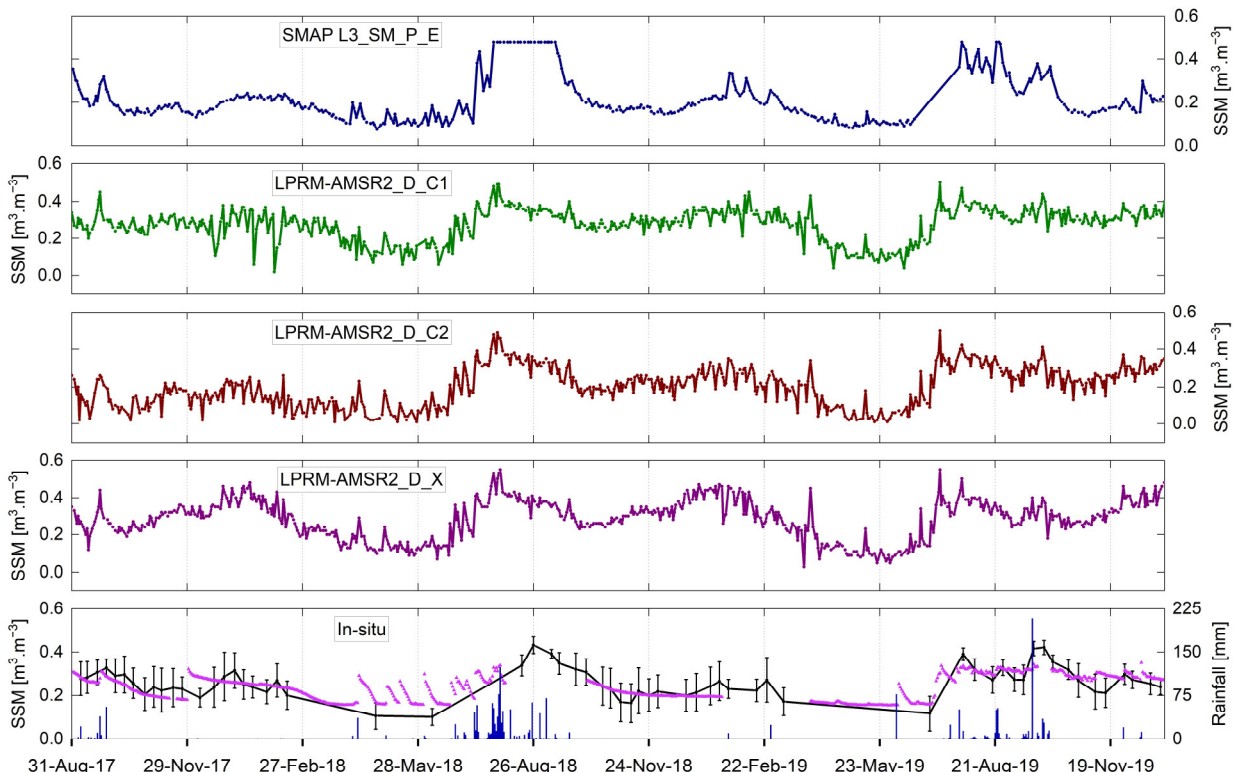

**Figure 5.** Timeseries of the soil moisture data derived from SMAP L3 and AMSR2 (C1-, C2-, and X-bands) for the study period over CZO. The in-situ data shown at the bottom contain the spatial mean soil moisture derived from ThetaProbe measurements (black line) and the spatial mean calibrated SM100 (pink triangle). The error bar in the in-situ data indicates ±1 standard deviation of the ThetaProbe measured values. Daily rainfall, obtained from the in-situ installed weather station is shown along with the in-situ soil moisture.

### 3.3. Comparison of Satellite and In-Situ Soil Moisture Observation

#### 3.3.1. Bias, RMSE, ubRMSE, and *R*

Figure 7 demonstrates the performance of all the soil moisture products under consideration. The computed values of bias, RMSE, ubRMSE, and R are shown along with the scatter plots. Since the SMAP and AMSR2 occupied a single footprint at the CZO, the comparison was conducted using the available timeseries information. It can be observed that SMAP-L3 and AMSR2-C2 had a dry bias of $-0.040$ and $-0.051$ m$^3$/m$^3$, respectively, whereas AMSR2-C1 and AMSR2-X showed a wet bias of $0.032$ and $0.041$ m$^3$/m$^3$, respectively. SMAP-L3 showed the lowest RMSE of $0.072$ m$^3$/m$^3$, while the RMSE appeared to be highest ($0.097$ m$^3$/m$^3$) for AMSR2-X. The performance regarding different soil moisture should be interpreted in a relative sense as there is a spatial mismatch between the satellite products, in addition to diurnal variation in point data collection and satellite retrieval. Both AMSR2-C1 and AMSR2-C2 produced the same correlation (R = 0.621) with the insitu data despite the contrast in overestimating and underestimating the in-situ values, respectively (Figure 7b,c). Furthermore, the SMAP-L3 data were found to show the best correlation (R = 0.712) with the in-situ measurements.

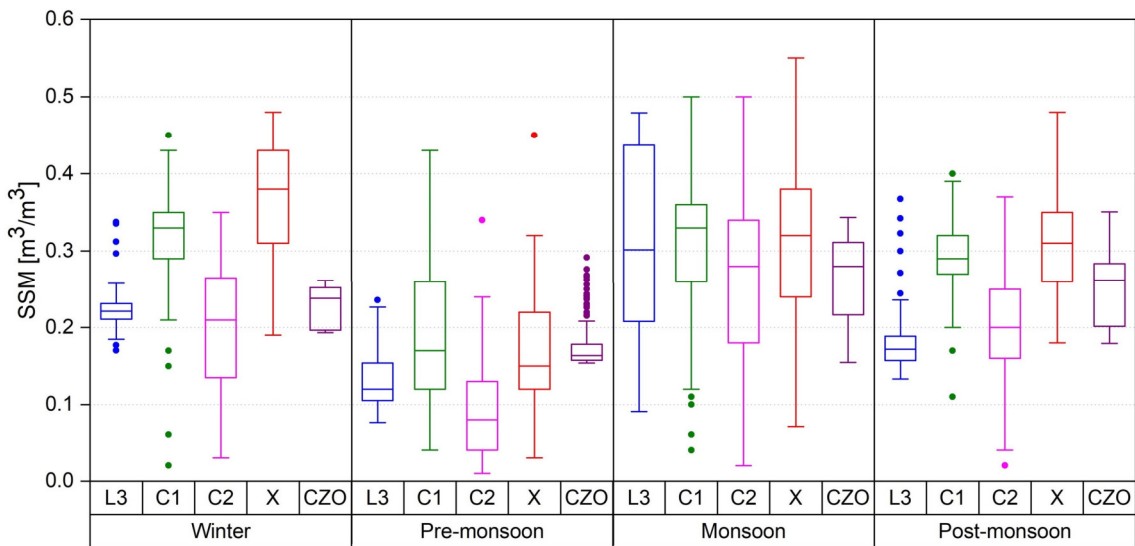

**Figure 6.** Box plot showing the range of seasonal variation of satellite and in-situ soil moisture over the CZO. The box boundary shows the interquartile range (25–75th percentile) along with the median (the soil line). The whisker length represents 1.5 times the interquartile range. The corresponding outliers are shown with filled solid circles. L3: SMAP_L3_SM_P_E, C1: LPRM-AMSR2_D_C1, C2: LPRM-AMSR2_D_C2, X: LPRM-AMSR2_D_X, CZO: mean in-situ value.

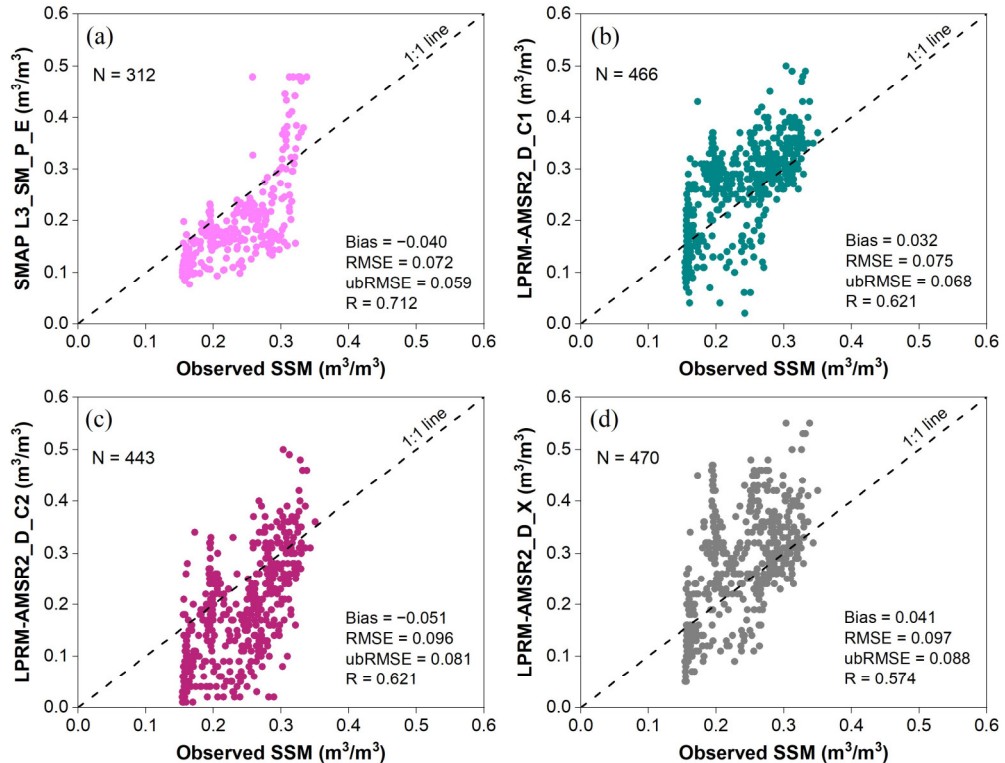

**Figure 7.** Scatter plot showing the daily observed and satellite grid soil moisture values. (**a**) SMAP-L3, (**b**) AMSR2-C1, (**c**) AMSR2-C2 and (**d**) AMSR2-X) for the study region. The 1:1 line is shown here as dashed lines, and the evaluation metrics (bias, RMSE, ubRMSE, and *R*) are shown for each comparison. The number of matched dates of the gap-free satellite pixel grid and the in-situ soil moisture campaign are shown, with the highest number of observations (N = 470) found for the X-band of AMSR2 data.

In addition to the overall comparison of the satellite products, the seasonal performance of individual product was assessed, considering the non-uniform rainfall variability and the irrigation strategies in north India. As shown in Figure 8, SMAP-L3 reflected the highest correlation among the remaining products for all seasons with a stronger representation during the monsoon period. The correlation coefficient (*R*) values were found to be 0.32, 0.33, 0.83, and 0.31 for the winter, pre-monsoon, monsoon, and post-monsoon periods, respectively. The highest correlation of SMAP-L3 during the wet period was also supported with zero bias for the same period. However, all SM products showed a significant increase in the *R*-value for the wet period compared to the remaining annual periods. Both the RMSE and the ubRMSE for SMAP-L3 showed the lowest value compared to the AMSR2 product during winter. While an extremely negative zero bias was observed for the AMSR2-C1 during the pre-monsoon period, a negative bias ranging from −0.1 to −0.03 m$^3$/m$^3$ was observed for the AMSR2-C2 SM product, including the seasonal variability throughout the year. It can also be seen that AMSR2-X had the worst performance in terms of annual seasonal variability, especially during winter when it had the largest bias, RMSE, and ubRMSE and lowest *R* of 0.14 m$^3$/m$^3$, 0.16 m$^3$/m$^3$, 0.08 m$^3$/m$^3$, and −0.09, respectively.

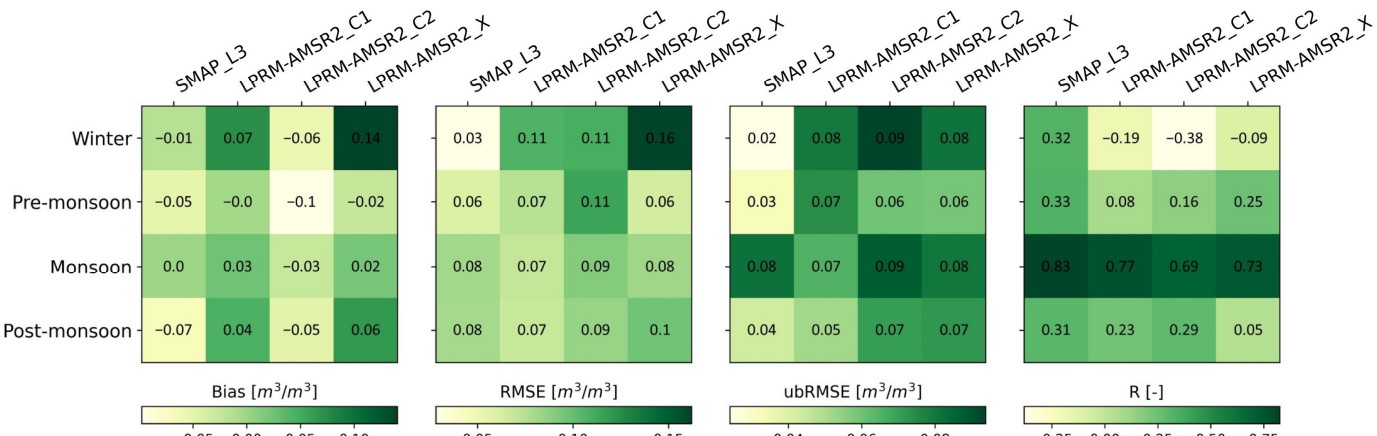

**Figure 8.** Seasonal (winter, pre-monsoon, monsoon, post-monsoon) evaluation of the four statistical parameters between the observed and satellite (SMAP and AMSR2) soil moisture products considered in this study over the CZO.

### 3.3.2. Evaluation of SMAP and AMSR2 Footprint through Taylor Diagram

Assessments of the geometrical relationships between different satellite SM products with the observed SM are presented through Taylor diagrams in Figure 9. The diagram integrally summarizes the fitness levels of statistical relations (R, RMSD, and standard deviation) between SMAP and AMSR2 products with the observed data over a two-dimensional polar plane. The closer a point is toward the observation point, the closer the satellite data are to the in-situ value. The diagram shows a negative correlation of AMSR2 products during the winter and pre-monsoon study periods, whereas SMAP-L3 was positively correlated throughout the study period. As observed in Figure 9e, SMAP-L3 showed a relatively higher average annual correlation (*R* = 0.7) and lower standard deviation (0.08 m$^3$·m$^{-3}$) and RMSD (0.06 m$^3$·m$^{-3}$), with the highest correlation (*R* = 0.83) during the monsoon period. The RMSDs of AMSR2 products (C1, C2, X) were observed to be 0.05–0.1 m$^3$·m$^{-3}$ throughout the seasonal and annual study periods. Overall, the SMAP-L3 performed best with respect to the observation value, followed by AMSR2-C1, C2 and X.

### 3.3.3. Error Variance for the Soil Moisture Triplets

We estimated the random error components for both satellite and ground soil moisture datasets using triple collocation (TC) analysis. The TC error was estimated by approximating the triplets for the SMAP, AMSR2, and SM100 soil moisture observation. Table 2 shows the number of possible triplets for the error estimation. Moreover, we considered

the SMAP (L3) for each triplet because of its high accuracy with respect to the ground observation as presented in the sections above. A total of 208 days of SM observations for SMAP, LPRM-AMSR2, and SM100 were considered to build the TC model. Figure 10 shows the TC error estimates for each dataset considered in this study. The error variances were within 4% for AMSR2 when the in-situ data were used as the reference. The error variance observed in the TC analysis also showed the best performance of SMAP-L3 with a value of $0.019 \text{ m}^3 \cdot \text{m}^{-3}$, followed by AMSR2-C2, SM100 (in-situ), AMSR2-C1, and AMSR2-X with values $0.032 \text{ m}^3 \cdot \text{m}^{-3}$, $0.034 \text{ m}^3 \cdot \text{m}^{-3}$, $0.035 \text{ m}^3 \cdot \text{m}^{-3}$, and $0.042 \text{ m}^3 \cdot \text{m}^{-3}$, respectively.

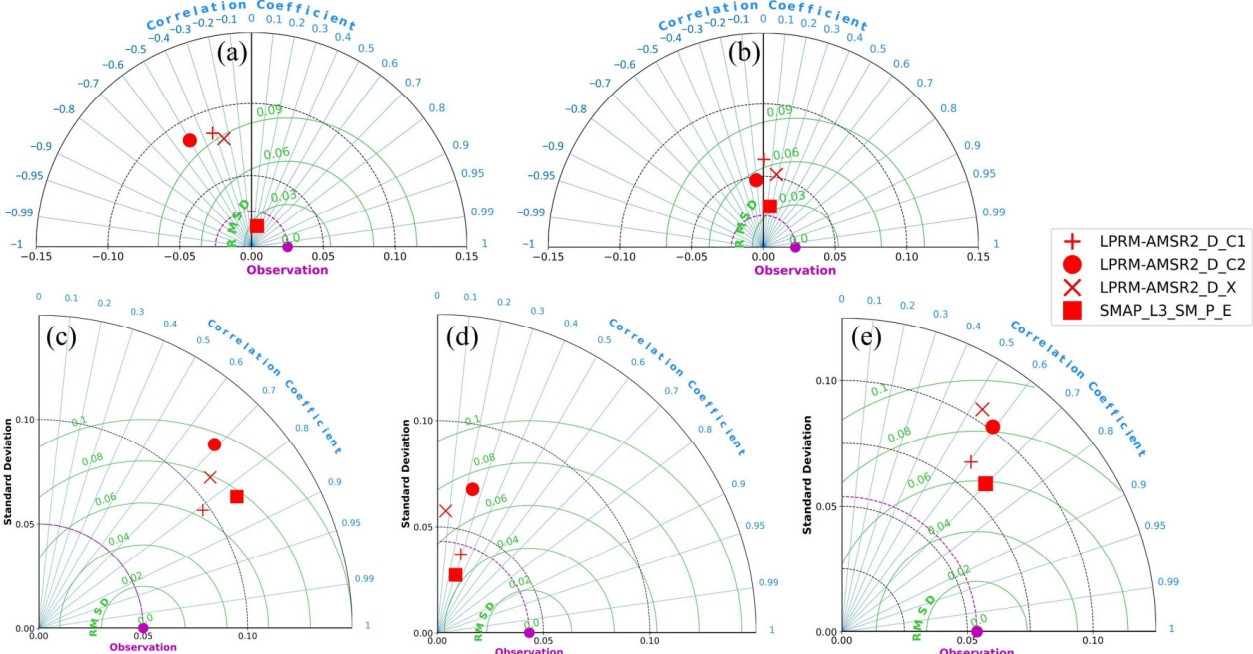

**Figure 9.** Taylor diagram illustrating the statistics of the comparison between SMAP and AMSR2 soil moisture products with in-situ data over CZO for (**a**) winter, (**b**) pre-monsoon (**c**) monsoon, (**d**) post-monsoon, and (**e**) all data. The dashed line shows the observed RMSE. Each marker in the graph shows the statistics of the corresponding soil moisture.

A high random noise in the AMSR2-X band was also reflected by the low SNR value of $-0.243$ dB (Figure 10). In contrast, SMAP-L3 showed more than 4-fold higher soil moisture timeseries signal contribution compared to random noise. In fact, the SNR of the L-band ranged from 5.78 dB to 7.48 dB with a mean of 6.63 dB. This suggests a clear advantage of L-band soil moisture timeseries over the C- and X-bands, where the signal was below 3 dB.

### 3.4. Soil Moisture Performance over Climatic Variables

The performance of the soil moisture simulations is predominantly influenced by precipitation and evaporation, while the evaporation is associated with temperature and radiation [74]. This motivated this study to establish the statistically significant relationships of the soil moisture datasets with major climatic variables (air temperature, daily rainfall, and potential evapotranspiration) using the nonparametric Spearman correlation coefficient. Figure 11 illustrates the monthly variation of ground-based observations of air temperature, daily rainfall, and $ET_0$ over the study region with significant correlation strength between in-situ and satellite soil moisture on a seasonal basis. In most seasons, the SM had a negative relationship with the 2 m air temperature. In fact, these relationships were significantly negative ($p < 0.05$) for the in-situ SM from January to September (winter to pre-monsoon period). The pre-monsoon and monsoon periods of the study region were significantly negatively correlated with air temperature for all remote and in-situ SM measurements. Interestingly, the LPRM-AMSR2 products showed a significant ($p < 0.05$) correlation with

the air temperature (2 m) for most seasons. Since surface soil temperature is significantly influenced by local air temperature, a possible explanation for this pronounced stronger negative correlation may be related to the retrieval of surface temperature by the LPRM algorithm. The LPRM retrieves the surface temperature from the available 36.5 GHz V-pol observations, whereas the SMAP surface temperature is predicted from the ancillary NASA GOES-5 model, and this may cause uncertainties to some extent in the SMAP surface temperature prediction [19]. Additionally, a greater influence of air temperature on the high-frequency observations might be partly contributed by passive microwaves (high frequencies) that are more attenuated by the vegetation layer, while the downward radiation from vegetation is reflected upward by the soil surface [32].

**Table 2.** Possible number of triplets for the SSM datasets. The first three triplets were considered in this study for the TC error analysis. SM100: in-situ soil moisture at CZO, L3: SMAP_L3_SM_P_E soil moisture, C1: LPRM-AMSR2_D_C1 soil moisture, C2: LPRM-AMSR2_D_C2 soil moisture, X: LPRM-AMSR2_D_X soil moisture.

|   | Triplets (Each Dataset Has a Length of 208) | | |
|---|---|---|---|
|   | *x* | *y* | *z* |
| 1 | SM100 | L3 | C1 |
| 2 | SM100 | L3 | C2 |
| 3 | SM100 | L3 | X |
| 4 | SM100 | C1 | C2 |
| 5 | SM100 | C1 | X |
| 6 | SM100 | C2 | X |
| 7 | L3 | C1 | C2 |
| 8 | L3 | C1 | X |
| 9 | L3 | C2 | X |
| 10 | C1 | C2 | X |

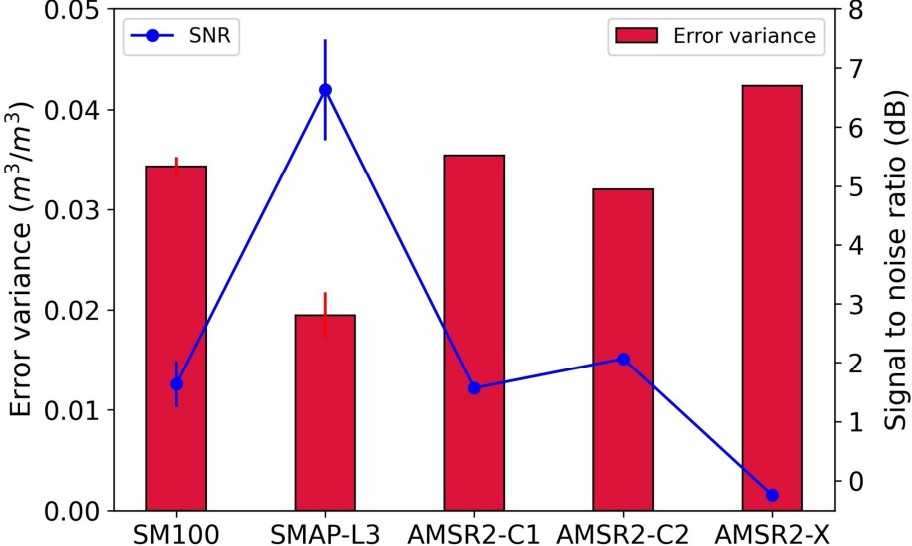

**Figure 10.** Estimated average error variances of the soil moisture products based on the triple collocation error model and the corresponding signal-to-noise ratio (SNR) for the remotely sensed and in-situ soil moisture observation in the study area.

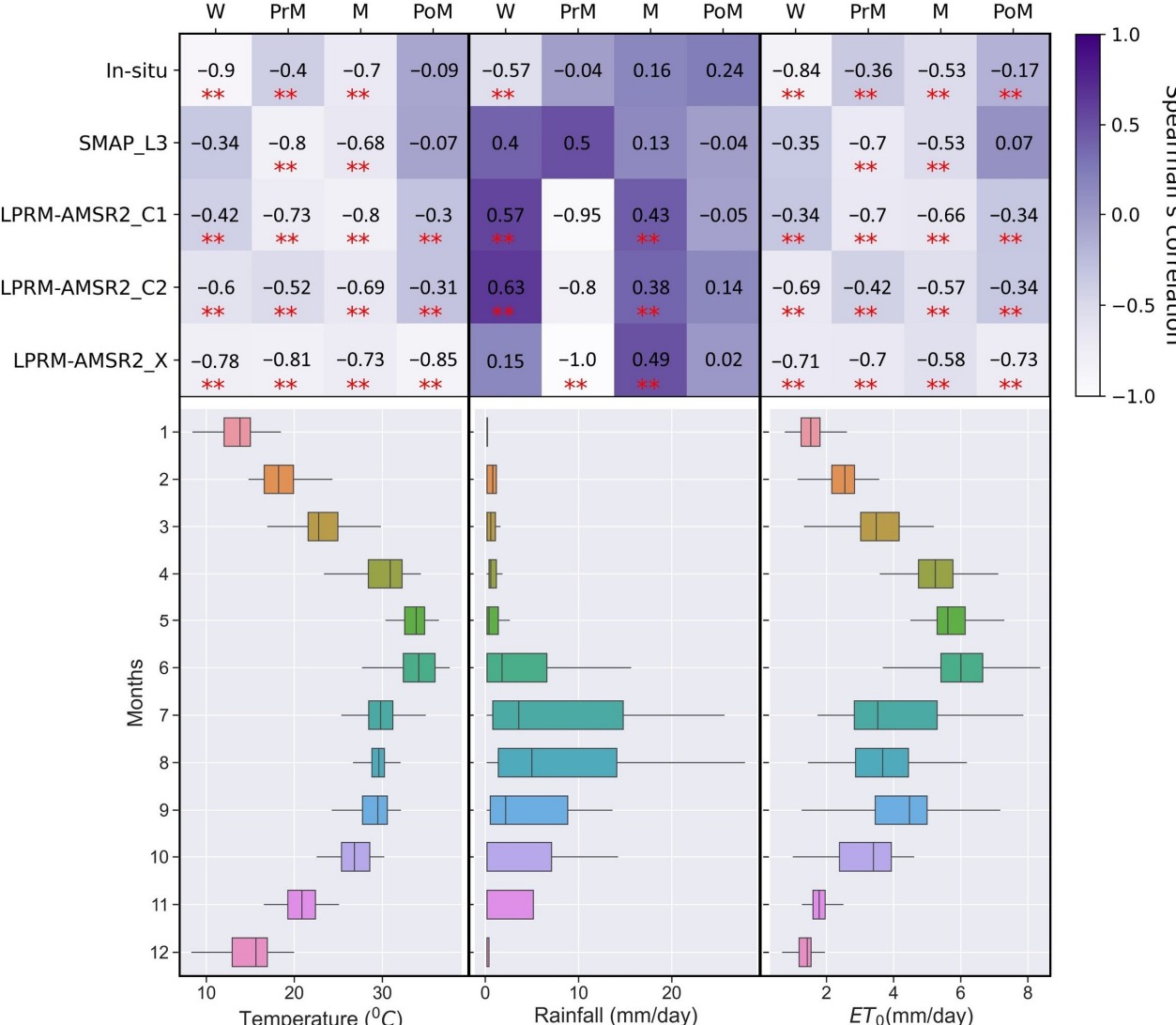

**Figure 11.** Top: Spearman's correlation coefficient between the daily soil moisture and the climatic variables (2 m air temperature, daily rainfall, and potential evapotranspiration (left to right)) for all seasons during the study period. W: winter, PrM: pre-monsoon, M: monsoon, and PoM: post-monsoon. Asterisks indicate statistically significant correlations (less than 5%). Bottom: monthly variation of air temperature, rainfall, and $ET_0$ (left to right) in the CZO with the box representing the interquartile range and the solid line within the box indicating the median value. The whisker length represents 1.5 times the interquartile range.

A significant negative correlation coefficient ($r = -0.34$ to $-0.73$ and $p < 0.05$) was also observed for the C- and X-band products with the estimated $ET_0$ for the study region, and this was true for all seasonal evaluations (Figure 11). Both in-situ and SMAP SM responded with a similar annual behavior toward the daily $ET_0$ and the air temperature variability. This was indicated by the negative and significant correlation coefficient of the concerned climate variables from the winter to monsoon period. Moreover, a significant relationship between ground-based SM and $ET_0$ was observed with a negative coefficient for all seasonal variability. The negative $ET_0$ forcing to SM accounted for the higher surface air temperature that considerably lowered the SM, and this was again reflected in the LPRM-derived SM, similar to its prominent relation with the air temperature.

Furthermore, we tried to relate the daily observed rainfall obtained from the in-situ weather station with the available soil moisture data, and the Spearman coefficient indicated a moderate to weak correlation between the two with a maximum *r*-value of 0.63 (Figure 11).

We also noted a negative relationship between SM observation and corresponding rainfall, and this relation was surprisingly seen for the in-situ sites for the winter and pre-monsoon periods. This can be attributed to the non-uniform distribution of rainfall variation in the study region, excluding the monsoon season. Moreover, the average $ET_0$ for CZO exceeds 1 mm/day during winter. Therefore, the early morning SM variation (considered in this study) may have contributed to the negative correlation towards the SM. A similar negative relation between precipitation and SM anomaly was reported in a recent study [22]. However, a marginally better positive correlation of in-situ SM with rainfall was observed after the pre-monsoon period, when the region experienced the maximum rainfall. The AMSR2 data also showed a strongly significant correlation with the daily rainfall variation at the 95% confidence level during the monsoon period.

### 3.5. Soil Moisture Relation to Vegetation Indices and Vegetation Optical Depth

Figure 12 shows the changes in vegetation indices and the VOD alongside the satellite and ground soil moisture observations over the study period in the CZO. Both satellite and ground SM observations responded very well to the temporal vegetation characteristics. Both 9 km and 10 km (resampled) EVI and NDVI were closely related to SM variation (Figure 12a). A closer analysis of the EVI and NDVI patterns suggests a periodic cropping pattern mostly followed in north India, specifically rice and wheat crops. Both NDVI and EVI had a similar variation in their amplitude and differed only by their magnitude; therefore, they were significantly correlated over the study area. However, we emphasize the relationship of EVI as it is the best representative of vegetation [21,75]. The peak of the periodicity in the EVI corresponded to the pre-harvesting stages of the major crops (rice and wheat). This periodic pattern of vegetation was also followed by soil moisture observations, acquired by the remotely sensed and ground-based platform in the study region throughout the study period.

A significant attenuation of the soil emitted energy occurs because of the presence of vegetation, which is represented by vegetation optical depth (VOD). Figure 12b presents the temporal evolution of the VOD for all evaluated microwave frequencies in the study domain. It shows that the LPRM had a much larger VOD than SMAP. The high VOD value also corresponds to low soil emissivity due to lower transmission of the vegetation. This ultimately corresponds to high SM content. This might be the possible reason for the higher SM values observed in Figure 12a corresponding to the high EVI and NDVI. Additionally, a smoother variation of the SMAP (SCA-V-pol) VOD was observed compared to DCA (showing a noisy pattern), simultaneously corresponding to the actual/satellite SM variability throughout the study period.

The Spearman correlation coefficients estimated between the timeseries of EVI and NDVI with the ground and satellite SM are presented in Table 3. The analysis shows that a strong and significant positive statistical relationship ($R$ = 0.75 to 0.79) existed between the SMAP and MODIS gridded data over the CZO. This strong correlation of the SMAP to the inherent SM may correspond to the V-pol SCA/Tau-Omega model that uses the vegetation water content climatology [76,77]. This observation is again supported in Table 4, where the SM had a significant and strong correlation with the SCA (V-pol) VOD. In addition, the EVI showed a more significant positive correlation with the in-situ observed soil moisture than NDVI. On the other hand, the C- and X-band data products did not show much correlation with the changes in the vegetation timeseries.

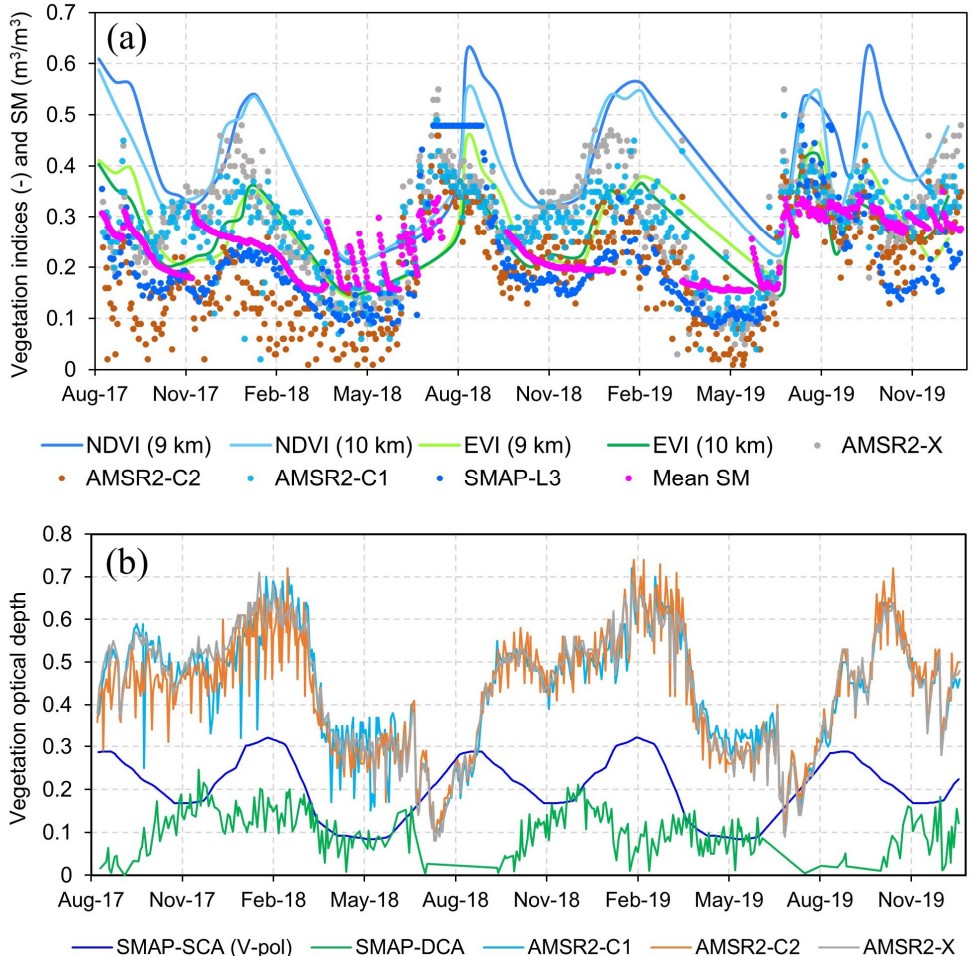

**Figure 12.** (**a**) Temporal evolution of MODIS-observed NDVI and EVI for the CZO with the satellite and ground soil moisture status. The NDVI and EVI were resampled to satellite soil moisture resolution (9 km and 10 km) for comparison. The single horizontal blue line around July–September, 2018 indicates the maximum SMAP soil moisture value for the observed soil type. (**b**) Vegetation optical depth timeseries of SMAP (SCA-V-pol and DCA) and AMSR2 (C1, C2, and X) for the study region.

**Table 3.** Spearman's correlation coefficient between daily soil moisture and vegetation indices (resampled).

|  | **NDVI (9 km)** | **NDVI (10 km)** | **EVI (9 km)** | **EVI (10 km)** |
|---|---|---|---|---|
| In-situ | 0.329 | 0.248 | 0.449 * | 0.432 * |
| SMAP_L3 | 0.789 * | 0.752 * | 0.775 * | 0.765 * |
| LPRM-AMSR2_C1 | 0.272 | 0.342 | 0.383 | 0.387 |
| LPRM-AMSR2_C2 | 0.085 | 0.238 | 0.179 | 0.255 |
| LPRM-AMSR2_X | 0.121 | 0.357 | 0.155 | 0.321 |

* Statistically significant correlations (less than 5%).

**Table 4.** Spearman's correlation coefficient between daily soil moisture and vegetation optical depth (VOD). DCA: dual-channel algorithm.

| | SMAP−SCA (V−pol) | SMAP (DCA) | VOD (C1) | VOD (C2) | VOD (X) |
|---|---|---|---|---|---|
| In-situ | 0.537 * | −0.142 | −0.037 | 0.067 | 0.018 |
| SMAP_L3 | 0.745 * | −0.019 | −0.099 | 0.028 | −0.016 |
| LPRM−AMSR2_C1 | 0.518 * | 0.081 | −0.014 | 0.064 | 0.008 |
| LPRM−AMSR2_C2 | 0.313 * | −0.052 | −0.302 * | −0.077 | −0.262 |
| LPRM−AMSR2_X | 0.418 * | 0.258 * | −0.077 | 0.086 | 0.025 |

* Statistically significant correlations (less than 5%).

## 4. Discussion

This study evaluated the microwave satellite soil moisture products (SMAP and AMSR2) using several statistical metrices and the TC error model over a tropical monsoon region of India. Although L-band (1.4 GHz) observations are optimal for soil moisture retrieval compared to higher frequencies such as C- and X-bands [7,18,19,78], we evaluated both frequency band products to depict the absolute overall performance. The study region, which is an extensively irrigated land, covers a single model grid of both satellite products with several in-situ monitoring locations. Therefore, we analyzed the daily temporal SM variability instead of both spatial and temporal domain coverage. Initially, we projected the continuous SM locations to predict the catchment mean using simple linear regression with multiple spatial observations. The utilization of a single monitoring location for regional SM variability through utilization of linear regression has been highlighted by several scientific and hydrological studies [64,66,79,80]. The projected catchment means of SM provide a good assessment of SM products of the presently operating passive microwave sensors along with their sensitivity to observed climate and vegetation dynamics on a seasonal and annual basis.

### 4.1. Overall Performance of SMAP-L3 and LPRM-AMSR2 Soil Moisture

Results on the evaluation of the SMAP-L3 and LPRM-AMSR2 satellite soil moisture data considered here demonstrated significant variability. The statistical metrices for daily variation showed that SMAP-L3 performed relatively well in predicting the ground-observed value (Figures 6 and 7a). Moreover, the L-band (1.41 GHz) product from SMAP was observed to have a small dry bias of $-0.04$ m$^3 \cdot$m$^{-3}$ toward the observed SM, in accordance with earlier work [36]. Although very few studies have reported the temporal dynamics of a single spatial footprint [21,23], the statistical evaluation of SMAP-L3 in this work yielded satisfactory results (ubRMSE = 0.059 m$^3 \cdot$m$^{-3}$), consistent with those from previous research on the Tibetan Plateau [28,29] and eastern Indian regions [50]. Although the overall evaluation of SMAP-L3 slightly disagreed with the accuracy requirement (0.04 m$^3 \cdot$m$^{-3}$) [56], the evaluation of the daily SMAP-L3 products provided a better estimate of the actual SM with the ubRMSE ranging from 0.024 m$^3 \cdot$m$^{-3}$ to 0.043 m$^3 \cdot$m$^{-3}$ excluding the monsoon period, as the transition from summer to monsoon season might significantly impact the signal accuracy (Figure 8). While the LPRM-AMSR2 products followed a similar pattern in terms of overall annual variability in the study region (Figure 5), the SM retrieval at 6.9 GHz performed better than that a 7.3 GHz and 10.7 GHz, with a wet bias of 0.032 m$^3 \cdot$m$^{-3}$ and ubRMSE of 0.068 m$^3 \cdot$m$^{-3}$ (Figure 7b–d). The biased estimates of the AMSR2 C- and X-band products were lower than the reported values for the Genhe region of China [81]. The overall variation of ubRMSE for the LPRM-AMSR2 ranged from 0.068 m$^3 \cdot$m$^{-3}$ to 0.088 m$^3 \cdot$m$^{-3}$, in accordance with earlier findings [35]. This also produced a relatively modest accuracy with respect to the desired accuracy goal (0.06 m$^3 \cdot$m$^{-3}$) for the AMSR2 mission objective.

Figure 13 further demonstrates the performance of the SMAP and AMSR2 products in terms of agreement with the actual soil moisture. Both L- and C-band soil moisture products

approached the actual soil moisture values if the in-situ soil moisture increased. However, the X-band product tends to overestimate (Figure 13d), which increased for higher in-situ measurements, in accordance with earlier findings [24,36]. A general overestimation of the LPRM SM product has been reported by previous studies [19,34,82], possibly because of the higher VOD, as shown in Figure 12b, and more attenuation [32]. Furthermore, the study region has a seasonal precipitation variation, with most rainfall occurring between the onset of monsoon season (June) and September. Since the range of the LPRM product is 0–1 $m^3 \cdot m^{-3}$, this causes a surplus soil water estimation exceeding the actual field capacity [34]. This might also be attributed to the low SNR for higher-frequency microwave bands caused by prominent cloud cover. The vegetation intensity during the monsoon period is also maximum, which reduces the quality of SM retrieval, as well documented in various studies [20,83]. On average, the SMAP-L3 better estimated the actual soil moisture although it had a small dry bias for the low in-situ SM, and the SMAP approached a minimal difference with the in-situ values upon approaching higher values (Figure 13a). This observation was prominent, especially during the monsoon period when the seasonal bias became zero due to frequent rainfall (Figure 8).

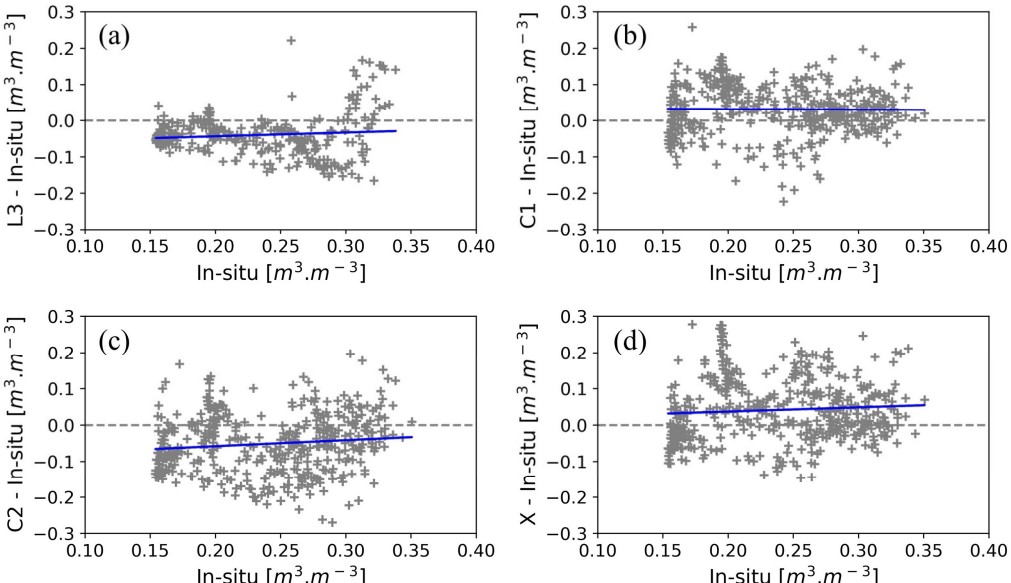

**Figure 13.** Observed distribution of the SMAP and LPRM-AMSR2 soil moisture products in the CZO, plotted as the difference between in-situ observations and (**a**) SMAP-L3, (**b**) AMSR2-C1, (**c**) AMSR2-C2, and (**d**) AMSR2-X products. The blue solid lines show the linear regressions.

The correlation among the satellite SM products considered here seemed to decrease from L- (SMAP) to X-band (AMSR2) microwave observations (Figure 7a–d). However, the C-band LPRM-derived SM showed a similar performance in terms of the correlation with the in-situ observations at both frequencies (6.9 and 7.3 GHz), matching with the LPRM-based SM of AMSR-E observations for Europe [84]. The correlation of 0.62 obtained for both the C1- and the C2-band (Figure 7b,c) was very similar, with a slightly better performance of C1 over C2 in terms of other statistical metrics. These findings are consistent with earlier studies [33,78,85]. In contrast to the penetration of 10.7 GHz which is more impacted by vegetation compared to the lower frequencies (6.9 and 7.3 GHz), we found a better performance of the 10.7 GHz product than 7.3 GHz during the monsoon season (Figure 8) possibly because of the higher vegetation water content, in accordance with the findings for the Murrumbidgee River catchment in Australia [85]. However, all products showed higher R-values during the monsoon period compared to their overall performance. In particular, the SMAP-L3 product showed the highest correlation coefficient (*R* = 0.826) during this period. This observation is also revealed from the Taylor diagram shown in Figure 9, where the SMAP-L3 showed an overall good correlation with the in-situ

measurements and AMSR2-X showed a relatively poor performance for the study region during the evaluated temporal window. The placement of the LPRM product (Figure 9e) mostly coincided with the recent study for an Australian catchment [24]. The SMAP-L3 showed the highest correlation ($R > 0.8$) in the monsoon period (Figure 9c) similar to the Little Washita Watershed, US [19], which also experiences a sub-humid climate.

The validation analysis was further extended to estimate the variance of the random error component in the daily temporal SM datasets using the triple collocation (TC) technique (Figure 10). The TC-based estimates agreed with the statistical metrics, and the SMAP-L3 outperformed the remaining datasets with a higher soil moisture signal than error components. The SMAP-L3 error component found in this study closely matched earlier findings [47,86]. The AMSR2-C1 exhibited relatively more error (and lower SNR) than the AMSR2-C2 SM, in contrast to the overall accuracy when considering the RMSE and ubRMSE of these two products (Figure 7). This may be attributed to the difference in the number of days of sampling considered for each of the evaluation processes (overall and TC statistics). However, the TC estimates found for the AMSR2 SM products were well within the limit of reported values [25,86]. Earlier work [87] also showed that the average error ranged from 0.02 to 0.06 m$^3 \cdot$m$^{-3}$ as obtained from the TC analysis using the International Soil Moisture Network and additional data from two satellites (ERA Interim and CCI).

*4.2. Possible Sources of Error in the SMAP-L3 and LPRM-AMSR2 Observations*

Differences in the performance of the above datasets might have occurred due to the error coming from several sources during the retrieval mechanism and the targets under consideration. In addition to the associated retrieval algorithms of SMAP [54] and LPRM-AMSR2 [33], the uncertainties in the ancillary parameters (surface temperature, surface roughness, vegetation, soil texture) during the retrieval process are the major sources of error [29,75]. Among these, vegetation and soil surface roughness are the most influencing parameters for L-band retrieval, as highlighted by Neelam and Mohanty (2015) [88]. This section, therefore, discusses some of these parameters with respect to the region of interest.

Sampling depth between in-situ and satellite observations: A part of the error came from the lack of agreement in the sensing depth between the satellite and the in-situ SM observations. Many studies have reported the effective sampling depth of L-band and C/X-band observations to be 0–3 cm and ~1 cm, respectively [35,82], whereas the ground measurements compared here were obtained at 5 cm (SM100) and 6 cm (ThetaProbe) soil depth. Therefore, the in-situ observations may not reliably represent the actual SMAP and AMSR2 SM retrieval. In addition, there was a difference in daily measurement interval between the satellite and the calibrated ground observation data because of the sampling durations of the distributed point measurements. Furthermore, we assumed a similar change in the dielectric properties of soil–water within the temporal window during an individual sampling period.

Sensitivity to vegetation: The vegetation density or the VOD has a significant contribution to the performance when retrieving the remotely sensed volumetric water content, by attenuating the passing signal from the soil surface and by emitting its own radiation. We found a statistically significant positive correlation of SMAP-L3 and in-situ SM to EVI in the study region (Table 3) according to the application of climatological NDVI in SMAP soil moisture estimation. This observation was supported by the VOD (SCA-V-pol) in the temporal window, showing a significant and strong positive correlation with all SM observations, especially with the SMAP observation (Table 4). Furthermore, the LPRM observations showed relatively low *R*-values with the EVI, which is in line with earlier findings [34] stating that the correlation coefficient decreases after EVI > 0.3. In fact, we found that around 47% (resampled at 9 km) and 39% (resampled at 10 km) of the daily EVI timeseries were above 0.3 during the study window. The low *R*-value of the LPRM may correspond to the overestimation of the observations seen in this study, which resulted because of the large VOD compared to SMAP (Figure 12b), in line with previous work [19].

Surface roughness: The accuracy of SM retrieval is also influenced by the surface roughness characteristics of the target area. As the surface topography of CZO is primarily dominated by croplands, the soil surface is modified locally (Figure 14) before and after the crops are sown; this causes micro roughness for brightness temperature retrieval during microwave scattering. Since the regional coverage of the study region has a typical monsoonal cropping pattern, the locally developed surface roughness could play an important role in modifying the soil emissivity and the subsequent retrieval algorithm. Previous work [34] identified surface roughness as a primary factor affecting the retrieval algorithm of the LPRM product. A recent study on the effect of surface roughness on the soil moisture retrieval showed that the performance of SMAP-L3 decreased with increasing the roughness, whereas there was a slight increase in the performance of LPRM product at a higher roughness condition [36]. This might have been the reason for the relatively higher *R*-value (0.232 and 0.286) for LPRM C1/C2 performance and the lower *R*-value (0.305) for SMAP-L3 performance during the post-monsoon period (October–December) compared with the winter and pre-monsoon periods (Figure 8).

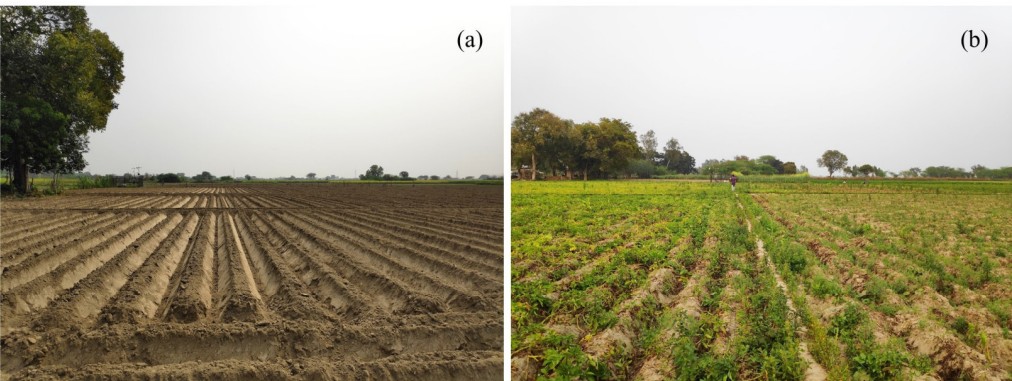

**Figure 14.** Field photos showing (**a**) field-scale micro roughness due to locally modified soil surface before the sowing period within CZO region during October, and (**b**) matured potato crops grown in the same area in February.

Radiometric contributions from water bodies: The brightness temperature ($T_b$) in the SM retrieval algorithm can be influenced by the radiometric characteristics of the water bodies within the corresponding pixels. Examination of the land cover of the study region showed numerous small water bodies including one large water body, along with two major canals for agricultural purposes. Even after the correction for the presence of the water bodies in the satellite retrieval algorithm, there may still be impacts on the $T_b$ values which can cause a warmer bias, resulting in a higher SM [18,89]. The signal can also be attenuated because of the temporal surface ponding effects in the paddy fields during intense rain events.

## 5. Conclusions

Assessment of the remotely sensed soil moisture information over several hydroclimatic regimes is crucial for their utilization in hydrological research and applications. This study comprehensively examined the grid scale L-band (SMAP) and C/X-band (LPRM-AMSR2) soil moisture (SM) product over the critical zone observatory (CZO) in the central Ganga plains, North India. Spatially calibrated long-term in-situ SM data from a continuous monitoring station were utilized to evaluate the microwave SM products through several statistical metrices and the triple collocation technique. This study also demonstrated the robustness of the ground and remotely sensed soil moisture observations to varied climatic and vegetation backgrounds under the observed seasonality and agricultural pattern over the study region.

Barring a slight dry bias ($-0.040$ m$^3 \cdot$m$^{-3}$), the SMAP-L3 product demonstrated a better performance compared to LPRM-AMSR2, correlating well with the in-situ measure-

ments. SMAP-L3 met the accuracy requirements of 0.04 $m^3 \cdot m^{-3}$ during most of the seasons (winter, pre-monsoon, and post-monsoon) with ubRMSE ranging from 0.024–0.043 $m^3 \cdot m^{-3}$. Within the LPRM-associated high-frequency datasets (C/X), the X-band product showed the lowest performance metrices compared to its lower frequencies, reducing the potential of the associated algorithm for further development. Furthermore, a stronger agreement of the current satellite products with the in-situ observations was found during the wet period (monsoon) compared to the remainder of the year. Additionally, triple collocation, applied to the datasets for the theoretical uncertainty estimation, showed a better performance of SMAP-L3 with the highest signal-to-noise ratio and an average error variance of $0.02 \pm 0.003$ $m^3 \cdot m^{-3}$ compared to the remaining datasets.

The sensitivity of the SM (in-situ and satellite) variability was assessed over the ground-based daily observed climatic variables such as precipitation, air temperature, and potential evapotranspiration ($ET_0$) and satellite-derived vegetation parameters such as vegetation optical depth (VOD) and vegetation indices using the Spearman correlation coefficient (*R*). The analysis revealed a statistically significant negative correlation for most of the SM datasets with the air temperature and $ET_0$, along with a satisfactory correlation for the observed precipitation pattern. It is worth noting that the LPRM-SM product exhibited a very good performance in response to the climatic attributes, underlining its importance as an independent climate data record. The correlation established between the SM and the vegetation indices, especially the EVI, showed a strong significant positive pattern with SMAP-L3, possibly because it uses the SCA (V-pol) vegetation climatology for soil moisture retrieval. On the other hand, the AMSR2 products revealed low and comparable *R*-values relative to the in-situ SM. Overall, the results and analysis presented in this study are helpful for understanding the quality of current satellite soil moisture products, highlighting their utility as an independent climate data record for agriculture and hydrologic applications.

**Author Contributions:** S.K.D. contributed in terms of conceptualization, methodology, formal analysis, and original draft preparation. R.S. contributed in terms of supervision, review and editing, resources, project administration, and funding acquisition. All authors have read and agreed to the published version of the manuscript.

**Funding:** This research was funded by the Ministry of Earth Sciences, New Delhi (grant number MOES/PAMC/H&C/63/2015-PC) for setting up the CZO.

**Institutional Review Board Statement:** Not applicable.

**Informed Consent Statement:** Not applicable.

**Data Availability Statement:** All remote sensing satellite data (SMAP, LPRM-AMSR2, and MODIS) used in this study are publicly available. The in-situ soil moisture and climatic datasets used in this study were derived from the ground-deployed sensor network at the HEART CZO and are available upon request from the corresponding author.

**Acknowledgments:** The first author acknowledges the PhD fellowship from IIT Kanpur. The authors also acknowledge the help from Shivam Tripathi and Surya Gupta during the installation of different sensors at the CZO, as well as discussions at various stages. The authors would like to thank the editor and three anonymous reviewers for their valuable comments and suggestions that improved the manuscript.

**Conflicts of Interest:** The authors declare no conflict of interest. The funders had no role in the design of the study; in the collection, analyses, or interpretation of data; in the writing of the manuscript, or in the decision to publish the results.

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
