# Peer review of "A Comprehensive Evaluation of Gridded L-, C-, and X-Band Microwave Soil Moisture Product over the CZO in the Central Ganga Plains, India"

_remotesensing, doi:10.3390/rs14071629_

Round 1
Reviewer 1 Report
The manuscript entitled “A comprehensive evaluation of gridded L, C, and X-band microwave soil moisture product over the Critical Zone Observatory (CZO) in the central Ganga plains, India using in-situ observations” evaluated SMAP-L3 and three LPRM AMSR-2 soil moisture (SM) products based on two years of data from one temporally continuous SM observation site. The single-site observations were calibrated to represent temporal SM variations over a 22-km2 Critical Zone Observatory (CZO) site, which falls approximately within a 9x9 km SMAP grid and a 10x10 AMSR2 grid. At the first glance, the audience are led to believe that the authors have installed 21 SM observation sites. Although the main conclusions might hold for most cases, the methodology is not quite right. I think that a linear transformation of single-site data cannot represent the gridded SM. Whether a linear transformation is applied, the R values are unchanged when the satellite products are validated against in-situ data. Figure 4 shows that the R2 value (0.5609) is low for the linear transformation, indicating that the SM spatial distribution cannot be captured (44% variance missing) by the single-site observations.
(1) Figure 3(b) uses two clusters of data for calibration. Obviously, the calibration is not good for either dry or wet soils.
(2) Figure 4(b) is unnecessary. When you perform linear regression and use the regression equation to recalculate y and then compare it with the original y, the slope value must be very close to unity and the bias value very close to zero…
(3) The authors tried to evaluate multiple SM products, and therefore the sections 3.4 and 3.5 should focus on comparison itself rather than disclose some patterns. What relationship is true? The one between potential variables (meteorological and biological) and in-situ SM or SMAP or AMSR2?
(4) Table 2 shows unexceptionally high R values between SMAP SM and vegetation indices. Is it because SMAP SCA use the climatological NDVI for SM retrieval? Is it possible to compare SMAP SCA and DCA VOD data for sure? The LPRM retrieves simultaneously SM and VOD. All AMSR2 SM data show low and comparable R values relative to in-situ SM.
(5) Line 628-629. The causes of SMAP SM biases are complicated. Figure 13(a) does not show that SM bias approaches 0.
(6) Line 668-669. The length of samples is 208 for all triplets (see Table 1). Also, I am not sure that two radiometer-based SM products are suitable to compose a triplet. ASCAT retrievals or model outputs are more acceptable.
(7) Line 463-464. Why the SD value is larger than the RMSD value?
(8) Line 189. Why using the nearest neighboring method? I think spatial aggregation is more relevant.
(9) Section 2.2.2. Be more specific on downscaled AMSR2 products, including the downscaling strategy and product usage in other studies.
(10) Line 152. Consistent soil and canopy temperatures.
(11) Line 138-139. Ascending at 6PM, and descending at 6AM.
(12) Line 100. The literature [43] (IEEE TGRS) is about paddy rice field.
Author Response
We thank the reviewer for his critical comments. We have addressed all comments from the reviewer, and our detailed response is attached herewith.

Reviewer 2 Report
The results indicate that the use of microwave-based input along with triple collocation (TC) and filtering is a viable and preferred alternative to the use of land surface models in soil moisture climate data records from dynamic response of soil moisture observations to climatic and vegetation parameters.
This paper will contribute from a tabular format presentation of an overview and characteristics of used passive microwave satellite sensors.
The discussion section is lacking a clarification of use or not on inter-calibration of data used.
The section 4.1 will benefit from a clearer reflection of the differences of SMAP-L3 and LPRM-AMSr2 Sm system used.
Also not clear if is a general better agreement of the SMAP/LPRM dynamics with respect to L-band and C/X band observations.
The conclusion section not clear in final part regarding a more standardized input for SM retrievals as compared to SM retrievals from higher frequencies with which it is merged, e.g., X-band from LPRM, and reduce the potential effects of land surface models on the data, strengthening its function as an independent climate data record.
Author Response

(The authors gave the same response as above.)

Reviewer 3 Report
Dear, Authors,
The proposed work is devoted to the validation of existing satellite soil moisture products using the Central Ganga plains, India as an example. The work carried out is an example of how such studies on sub-satellite validation of satellite products should be carried out.The article can be accepted in the form in which it is written.
Essentially, I have only one question. Of course, the moisture content of the soil surface varies greatly within a pixel of sounding from a few percent to tens of percent. The issue of spatial averaging in this work is well solved using 20 points distributed over the pixel area. The authors rightly point out that the depth of moisture sensing is in the range of less than 5 cm (actually 0-1, 0-2 cm) in L-band, and neven less it is in the in C- and X-band (soil surface <0.5 cm)! Why did the authors, realizing this, the very first sensor was installed at a depth of 5 cm, and not on the surface (vertically down with pins from the soil surface)!? (Some comments contains in attached file.)
Best regards.

Author Response

(The authors gave the same response as above.)

Round 2
Reviewer 1 Report
The authors have made moderate revisions and massive explanations in this version. But I still do not fully agree with the method used in this study. Linear regression is insufficient to transfer point measurements to grid averages.
Figure 3(b) lacks soil mositure data in dry-wet transition period, making it not a robust regression equation.
for the response (6), ASCAT has a coarser footprint???
fot the response (7), by definition SD is ubRMSE, why SD is larger than ubRMSE?